# Recognition of the small regulatory RNA RydC by the bacterial Hfq protein

Daniela Dimastrogiovanni[1], Kathrin S Fröhlich[2,3], Katarzyna J Bandyra[1], Heather A Bruce[1], Susann Hohensee[1], Jörg Vogel[3], Ben F Luisi[1]*

[1]Department of Biochemistry, University of Cambridge, Cambridge, United Kingdom; [2]Department of Molecular Biology, Princeton University, Princeton, United States; [3]Institute for Molecular Infection Biology, University of Würzburg, Würzburg, Germany

**Abstract** Bacterial small RNAs (sRNAs) are key elements of regulatory networks that modulate gene expression. The sRNA RydC of *Salmonella sp.* and *Escherichia coli* is an example of this class of riboregulators. Like many other sRNAs, RydC bears a 'seed' region that recognises specific transcripts through base-pairing, and its activities are facilitated by the RNA chaperone Hfq. The crystal structure of RydC in complex with *E. coli* Hfq at a 3.48 Å resolution illuminates how the protein interacts with and presents the sRNA for target recognition. Consolidating the protein–RNA complex is a host of distributed interactions mediated by the natively unstructured termini of Hfq. Based on the structure and other data, we propose a model for a dynamic effector complex comprising Hfq, small RNA, and the cognate mRNA target.

*For correspondence: bfl20@cam.ac.uk

**Competing interests:** The authors declare that no competing interests exist.

**Reviewing editor**: Gisela Storz, National Institute of Child Health and Human Development, United States

## Introduction

The expression of genetic information is controlled and synchronised through intricate regulatory networks. In bacteria, the control of gene expression post-transcription is mediated in part by small RNAs (sRNAs), and their contributions enrich the computational complexity and repertoire of regulatory circuits (*Beisel and Storz, 2011*). Bacterial sRNAs are typically 50 to 300 nucleotides in length (*Storz et al., 2011*), and each control expression of a distinct set of target mRNAs, which they recognise with high specificity and apparent precision. One of the major facilitators of sRNA activity in bacteria is the protein Hfq, which promotes pairing of an sRNA with its target mRNA in solution (*Møller et al., 2002*; *Vogel and Luisi, 2011*; *Panja and Woodson, 2012*; *De Lay et al., 2013*). Indeed, the kinetics of target pairing seem an essential aspect of sRNA action in vivo, as many of these sRNAs affect rates of translation or decay, either positively or negatively depending on target and context (*Storz et al., 2004*; *Fröhlich and Vogel, 2009*; *Desnoyers and Massé, 2012*; *Papenfort et al., 2013*). It can be envisaged how Hfq acts as a catalyst for such recognition in vivo, but the question naturally arises how sRNAs generally achieve precision, accuracy, and speed in producing their effects and avoid undesired off-target consequences. The key to understanding these fundamental processes of sRNA-based regulation is to determine how RNAs are bound and presented by Hfq and other effector proteins.

The previous crystal structures of truncated Hfq variants have provided clues as to how the protein recognises short stretches of single-stranded RNA. Hfq bears an α + β fold that is the signature of the highly conserved family of Sm/Lsm proteins that draws members from all domains of life (*Kambach et al., 1999*; *Wilusz and Wilusz, 2013*). Like other proteins of this extensive group, the bacterial Hfq self-assembles into a ring-like architecture. Hfq of *Escherichia coli* and other eubacteria forms a compact hexameric toroid that presents two structurally non-equivalent concave surfaces for molecular recognition; these faces are referred to as the proximal and distal faces, with the former exposing an N-terminal α-helix (*Schumacher et al., 2002*; *Link et al., 2009*). Crystallographic studies have identified

**eLife digest** A crucial step in the production of proteins is the translation of messenger RNA molecules. Other RNA molecules called small RNAs are also involved in this process: these small RNAs bind to the messenger RNA molecules to either increase or decrease the production of proteins.

Bacteria and other microorganisms use small RNA molecules to help them respond to stress conditions and to changes in their environment, such as fluctuations in temperature or the availability of nutrients. The ability to rapidly adapt to these changes enables bacteria to withstand harmful conditions and to make efficient use of resources available to them.

Many small RNA molecules use a protein called Hfq to help them interact with their target messenger RNAs. In some cases this protein protects the small RNA molecules when they are not bound to their targets. Hfq also helps the small RNA to bind to the messenger RNA, and then recruits other enzymes that eventually degrade the complex formed by the different RNA molecules.

Previous research has shown that six Hfq subunits combine to form a ring-shaped structure and has also provided some clues about the way in which Hfq can recognise a short stretch of a small RNA molecule, but the precise details of the interaction between them are not fully understood. Now Dimastrogiovanni et al. have used a technique called X-ray crystallography to visualize the interaction between Hfq and a small RNA molecule called RydC.

These experiments reveal that a particular region of RydC adopts a structure known as a pseudoknot and that this structure is critical for the interactions between the RydC molecules and the Hfq ring. Dimastrogiovanni et al. find that one RydC molecule interacts with one Hfq ring, and they identify the contact points between the RydC molecule and different regions of the Hfq ring.

Based on this information, Dimastrogiovanni et al. propose a model for how the RydC:Hfq complex is likely to interact with a messenger RNA molecule. The next step will be to test this model in experiments.

interactions of short RNA polymers with either of these two surfaces and have inferred sequence preferences that have been corroborated and refined by mutagenesis and solution binding studies (*Schumacher et al., 2002*; *Link et al., 2009*; *Sauer and Weichenrieder, 2011*; *Robinson et al., 2013*; *Peng et al., 2014*). According to these results, the proximal site interacts preferentially with uridine-rich sequences, while the distal site favours the binding of ARN or ARNN motifs (with R being a purine and N any nucleotide) (*Schumacher et al., 2002*; *Link et al., 2009*; *Sauer and Weichenrieder, 2011*). In addition to the distal and proximal faces, the torus-shaped Hfq hexamer bears a convex circumferential rim that has recently been identified as a surface contributing to RNA binding (*Sauer et al., 2012*; *Zhang et al., 2013*). Structural studies of the hetero-heptameric Sm assembly from the mammalian splicing machinery have identified RNA binding surfaces that share some similarities with the bacterial counterpart (*Leung et al., 2011*).

Bacterial sRNAs may recognize their cognate target mRNAs in different ways. In some sRNAs, a target-recognition region is presented at or near the 5'-end (*Papenfort et al., 2010*). For another subset of sRNAs, several pairing regions are non-continuous and each can independently bind distinct targets (*Beisel and Storz, 2011*; *Shao et al., 2013*). In both cases, the sRNA recognition site (referred to as the 'seed') and the target can have imperfect complementarity of various lengths with interactions being as short as 6 base-pairs, as seen for example in the SgrS/*ptsG* mRNA pair from *E. coli* (*Kawamoto et al., 2006*). To fully understand the mechanism of sRNA mediated regulation, it is important to address the questions of how the intricate RNA folds are recognised, how the seeds are presented, and how the pairing of sRNAs with mRNAs is facilitated. As the sRNAs, mRNAs, and their complexes can all be remodelled upon binding Hfq, the rules for recognition are likely to be highly dependent on context.

In this study, we describe the structure of the sRNA RydC in complex with the full-length Hfq protein of *E. coli*. Previous studies demonstrated that RydC is involved in biofilm regulation (*Bordeau and Felden, 2014*) as well as in the control of membrane stability through the positive regulation of

an isoform of the *cfa* mRNA encoding cyclopropane fatty acid synthase (*Fröhlich et al., 2013*). RydC is proposed to have a pseudoknot fold that exposes a 5' seed sequence, and the in vivo stability of RydC requires Hfq (*Antal et al., 2005*; *Fröhlich et al., 2013*; *Bordeau and Felden, 2014*). The crystal structure of the RydC–Hfq complex, together with biochemical and in vivo data, defines key interactions and suggests a hypothetical model for how the sRNA and mRNA targets might be matched through Hfq binding.

## Results

### RydC interaction with Hfq and pseudoknot structure is essential for its stability in vivo

Despite its low abundance in comparison with other sRNAs under standard growth conditions, RydC has been repeatedly recovered with the Hfq protein in pull-down assays (*Zhang et al., 2003*; *Sittka et al., 2008*; *Chao et al., 2012*). A distinctive feature of RydC is its predicted pseudoknot fold which is highly conserved and whose disruption by point mutations renders the RNA unstable in vivo (*Fröhlich et al., 2013*). A double mutant of RydC (G37C and G39C within helix 1; hereafter, RydC-S1) was predicted computationally to form a distinct structure with one short stem-loop at the 5'-end and a strong terminator hairpin at the 3'-end (*Figure 1A*).

To assess the molecular mechanism underlying the intrinsic instability of the RydC-S1 mutant, we compared its turnover rate to the wild-type RNA in the presence and absence of the chaperone Hfq (*Figure 1B/C*). To this end, both RydC and RydC-S1 were expressed in *Salmonella* under the control of the constitutive $P_L$ promoter from high-copy plasmids. At exponential growth ($OD_{600}$ of 1),

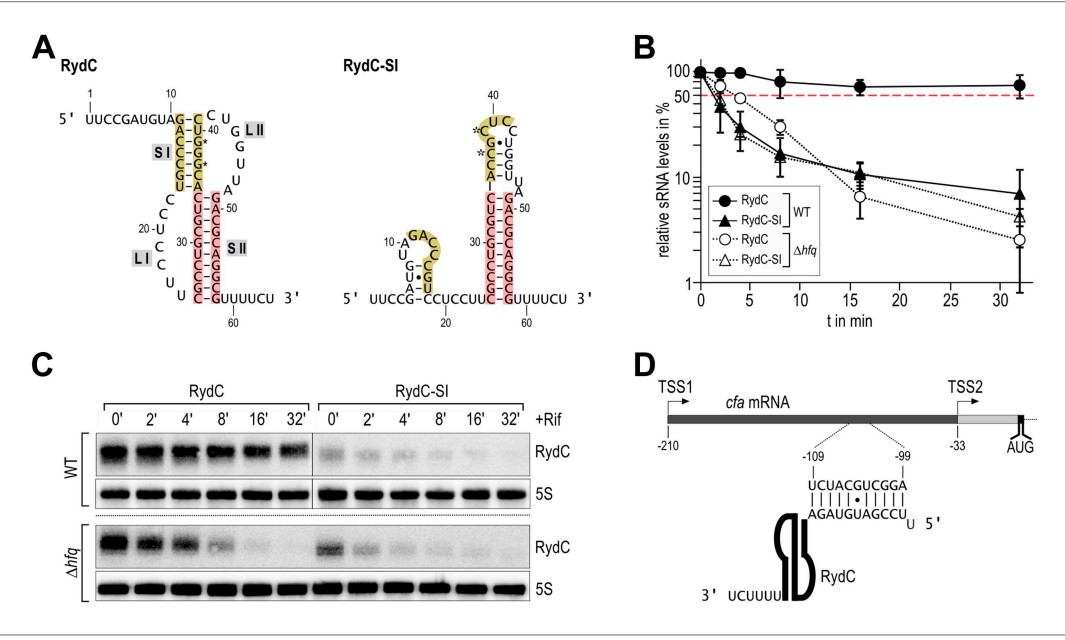

**Figure 1**. Association of RydC with the chaperone Hfq and its pseudoknot fold are required for sRNA stability. (**A**) Secondary structure of the RydC pseudoknot and the predicted alternative structure are formed by the double mutant RydC-SI. Substitutions G37C and G39C are indicated with asterisks. (**B**, **C**) Stabilities of RydC and RydC-SI were determined by Northern blot analyses. Total RNA samples were extracted prior to and at indicated time-points after inhibition of transcription by rifampicin in late exponential phase ($OD_{600}$ of 1) from *Salmonella* strains Δ*rydC* (JVS-0291) or Δ*hfq* (JVS-0584), carrying plasmids expressing RydC (pKF42-1) or RydC-SI (pKF60-1) from the constitutive $P_L$ promoter. Error bars represent standard deviation calculated from three biological replicates. (**D**) Predicted duplex formed between the RydC 5'-end (nts 2–11) and the longer isoform *cfa* mRNA (nts −99 to −109 relative to the translational start site) originating at transcriptional start site 1 (TSS1).

The following figure supplement is available for figure 1:

**Figure supplement 1**. Electrophoretic mobility shift assay (EMSA) of Hfq with RydC or RydC-S1.

transcription was stopped by the addition of rifampicin, and RNA levels were monitored at various time-points after treatment. As previously shown (*Fröhlich et al., 2013*), RydC is very stable in wild-type cells (half-life ($t_{1/2}$) >32 min), while RydC-S1 decayed rapidly ($t_{1/2}$ ~2 min). In contrast, the half-life of RydC is already markedly reduced in *hfq* mutant cells ($t_{1/2}$ ~4 min) and there is some further acceleration of the decay rate for RydC-S1, matching that of RydC in the absence of Hfq. This observation led us to assume that RydC stability depends on both the integrity of the pseudoknot fold and the ability to associate with the Hfq protein. Both the wild-type RydC and RydC-S1 mutant bind Hfq in electrophoretic mobility shift assays with in vitro synthesized RNA although the affinity of the mutant is lower than for the wild-type RydC (*Figure 1—figure supplement 1A*). Competition experiments suggest that the binding sites for the RydC and RydC-S1 overlap at least partially (*Figure 1—figure supplement 1B*). This suggests that the destabilisation in vivo might result mostly from the integrity of the pseudoknot fold, but to some degree, the Hfq binding alone may inhibit the degradation of RydC from initiating in single-stranded regions as suggested for other sRNAs (*Moll et al., 2003*; *Saramago et al., 2014*).

## Overview of the structure of the RydC–Hfq complex

The strong association of RydC with Hfq both in vivo and in vitro as well as its highly compact structure made it an ideal candidate for co-crystallizations. Co-crystals of full length *E. coli* Hfq and *Salmonella* RydC were obtained that diffracted to 3.48 Å resolution, and the crystal structure of the RydC–Hfq complex was solved by molecular replacement using a model of the structured hexameric core of *E. coli* Hfq (i.e., lacking the C-terminal residues beyond amino acid 65). Unbiased, interpretable density for the RNA was apparent in the early maps calculated from the positioned protein hexamer, and a model for most of RydC could be fitted into the electron density (*Figure 2—figure supplement 1* shows a map in which the RNA was omitted from the refinement and phase calculations). As the crystal structure is limited to a resolution of 3.48 Å, the map does not provide unequivocal alignment of RydC sequence with structure; however, the path of the majority of the RNA and the duplex regions could be modelled confidently. The asymmetric unit of the crystal comprises one full Hfq hexamer and one RydC (*Figure 2*). The fold of the RydC is consistent with the pseudoknot structure predicted from solution studies and sequence alignments (*Antal et al., 2005*; *Fröhlich et al., 2013*). The RNA bridges two adjacent Hfq hexamers in the crystal lattice, effectively forming a distorted sandwich with a wedge-shape (*Figure 3A*). The 3'-end of the RNA interacts with the proximal face of the principle hexamer, while the 5'-end interacts with the upper portion of the lateral surface of another hexamer in the asymmetric unit (see below for details, paragraph '*Interactions of RydC with the lateral surface of Hfq*').

The organisation of the RNA in the crystal would suggest that a single RydC molecule could form a closed complex with two sandwiching Hfq hexamers. However, solution studies using analytical ultracentrifugation (AUC) and multi-angle laser light scattering (SEC-MALS) indicate that the RydC can form a stable 1:1 complex with the Hfq hexamer under various buffer conditions when the components are present in unitary stoichiometry (*Figure 2—figure supplements 2–3*). When Hfq is present at twice the concentration of RydC under low ionic strength conditions, some 2:1 Hfq:RydC species can be identified; moreover, a higher order complex can be observed by electrophoretic mobility shifts when Hfq is in excess over RNA (results not shown). These observations suggest that the stoichiometry of Hfq/RNA complexes is very sensitive to solution conditions. In vivo, it seems unlikely that one RydC can find two Hfq hexamers when the protein is limiting to total RNA (*Wagner, 2013*), and we suggest that the 1:1 complex is likely to represent the physiologically relevant species. We propose that the 2:1 sandwiching complex we observe in the crystal is not the biologically relevant species, but that it encompasses the full set of interactions that would form in a closed 1:1 RydC:Hfq complex. We suggest that the contacts made by RydC with the neighbouring Hfq hexamers can form on the proximal face of the same, single hexamer to form an idealised 1:1 complex. This hypothetical model will be elaborated on further below.

## Interactions of the U-rich tail of RydC with the proximal face of Hfq

The 3'-end of RydC has a U-rich tail that - together with a strong hairpin structure - facilitates Rho-independent transcription termination and which is an important determinant for recognition of many sRNAs (*Otaka et al., 2011*; *Sauer and Weichenrieder, 2011*; *Ishikawa et al., 2012*). The RydC/Hfq structure reveals that the U-rich tail is bound in a recessed channel on the proximal face of Hfq. The electron density is not well defined in this region, and there may be disorder or

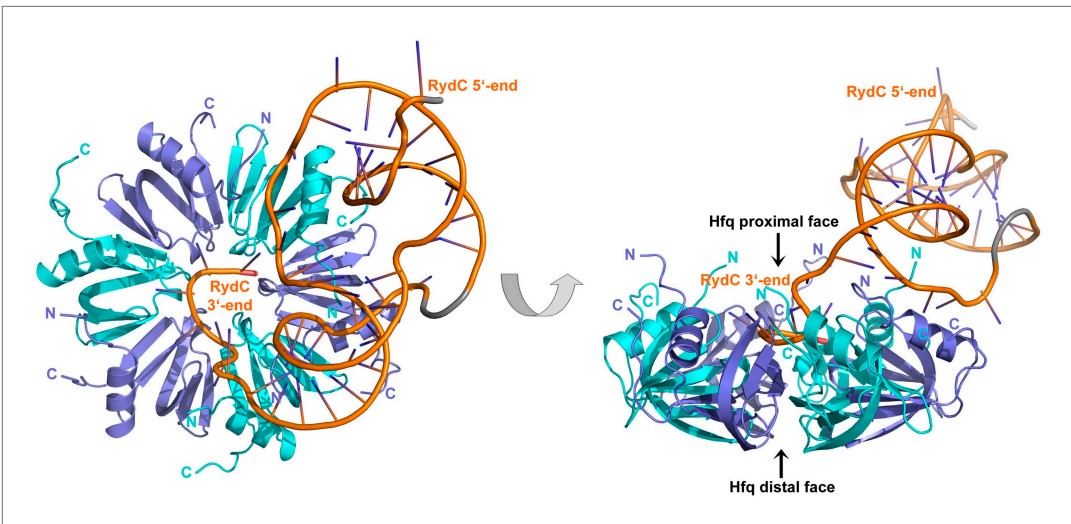

**Figure 2**. Crystal structure of the RydC/Hfq complex. The content of the asymmetric unit is shown. The left panel presents a view along the molecular sixfold axis of the Hfq ring core (roughly encompassing residues 3–72), while the right panel is viewed along the perpendicular direction. The 3′-end poly-U tail of RydC is inserted in a groove on the proximal face of the hexamer. The N- and C-terminal regions are indicated with letters (cyan and blue for different Hfq protomers); these are natively unstructured and in multiple conformations, but some portions could be modelled. The RydC phosphate backbone is shown as an orange cartoon, bases are shown as sticks. Nucleotides 20–21 are disordered, but the backbone was modelled to help view the RNA structure and is indicated in grey cartoon style, as is also the last modelled phosphate at the 5′-end of RydC.

The following figure supplements are available for figure 2:

**Figure supplement 1**. Electron density calculated after refinement of a model in which a portion of a RNA duplex region was omitted (nucleotides 26–33 and 50–57).

**Figure supplement 2**. Analytical Ultracentrifugation of Hfq, RydC, and Hfq–RydC mixtures at different RNA:protein ratios.

**Figure supplement 3**. Size exclusion chromatography dynamic light scattering (SEC-MALS).

multiple conformations of the polyU tail, but the features do show that uridines U61, U62, and U64 (as well as C63) each forms an aromatic base stacking interaction with the ring of F42 (*Figure 3C*; distances provided in *Figure 3—figure supplement 1*). K56 forms hydrogen bonds with the O2 group of the uracils, consistent with the importance of this residue in binding RNA on the proximal face (*Mikulecky et al., 2004*). The model indicates that there are likely to be hydrogen bonds between the uridine bases and Q41 and Q8, as well as between the furanose 2′ OH group with the imidazole group of H57. These interactions are consistent with the crystal structure of *Salmonella* Typhimurium Hfq/polyU (*Sauer and Weichenrieder, 2011*; PDB code 2YLC) and with fluorescence quenching experiments for binding $U_6$ (*Robinson et al., 2013*). Similar interactions are seen in the crystal structure of *Staphylococcus aureus* Hfq in complex with AUUUUUG (*Schumacher et al., 2002*; PDB code 1KQ2) and *E. coli* Hfq in complex with AUUUUUA (*Wang et al., 2011*, *2013*; PDB code 4HT9). In principle, the RNA can visit each of the uridine pockets following a clockwise or counter-clockwise path, but the polarity seems to be consistently clockwise in the reference frame that views the proximal face. The RNA exits the groove to engage on the rim of the proximal face of Hfq through numerous interactions with protein side- and main-chain atoms, as we describe further below.

## Interactions of RydC with the lateral surface of Hfq
On the exposed rim of the proximal face, U24 stacks onto F39 and U23 packs onto U24 (*Figure 4A*). The U46/U47 bases form similar interactions with the rim of Hfq in a neighbouring asymmetric unit

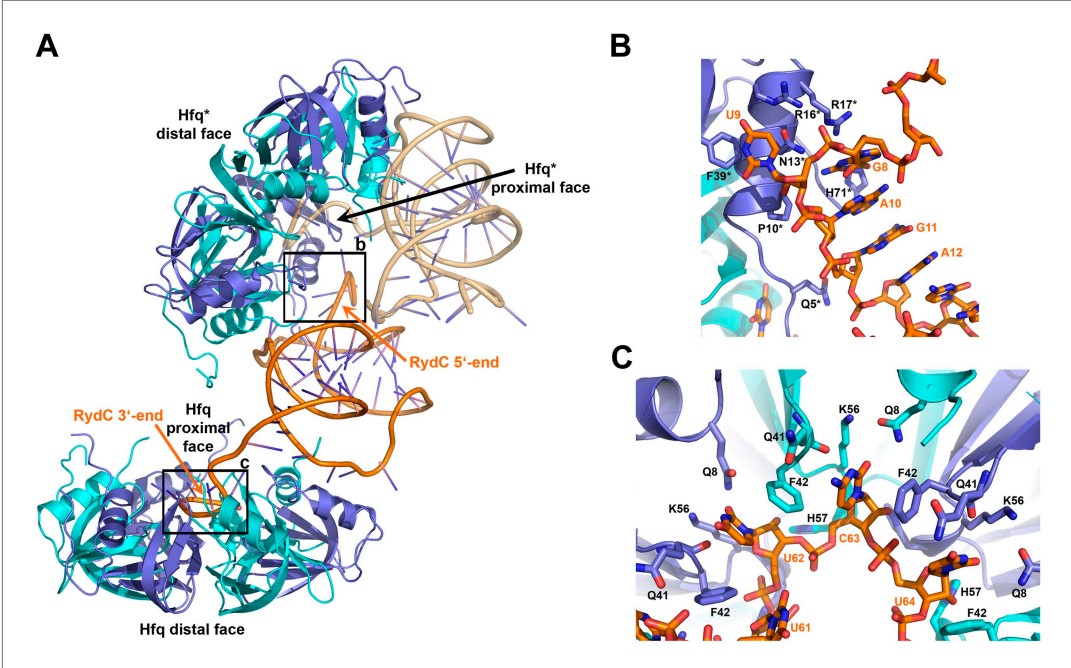

**Figure 3**. Interactions established by the 3'-end and 5'-end regions of RydC with two adjacent Hfq hexamers in the crystal lattice. (**A**) RydC is sandwiched between two Hfq hexamers in the crystal lattice. Two asymmetric units are shown with the RydC in the centre (coloured in orange) bridging two Hfq hexamers. The symmetry-related Hfq molecule is labelled with an asterisk (*). The 3'-end region of RydC in the main asymmetric unit is pointed by an arrow and contacts the proximal face of the principal Hfq hexamer, another arrow indicates the position of the 5'-end of RydC, interacting with a portion of the rim of the Hfq hexamer in the neighbouring asymmetric unit. Two black rectangles define the 5'-end and 3'-end RydC–Hfq contact regions, which are enlarged in panels **B** and **C** respectively. (**B**) Contacts established between the 5'-end seed region of RydC and the rim of a symmetry related Hfq molecule. The main interactions at this interface involve the following amino acid–nucleotide pairs: R17-G8, H71-G8, R16-U9, N13-U9, F39-U9, Q5-G11, P10-A10. Residues belonging to the symmetry related Hfq molecules are labelled with an asterisk (*). (**C**) Interactions of the poly-U 3'-end of RydC with the recessed groove of the proximal face of Hfq. The uridines and cytosine stack on F42; this and other interactions are similar to those seen in the U6/Hfq crystal structure (**Sauer and Weichenrieder, 2011**). The side chains of critical residues that contribute to keep RydC in the channel are shown in stick representation. The adjacent protomers of Hfq are coloured in blue and cyan, RydC is shown in orange. Side chains are shown for all residues, main chains are shown for Q41 in order to highlight the position of atoms involved in interactions with neighbouring residues. The RNA used in the crystallisations has a guanine on the 3'-end originating from the template for in vitro transcription, but this base may be in multiple conformations.

The following figure supplement is available for figure 3:

**Figure supplement 1**. Distances of the main contacts between Hfq and the 5'- and 3'-ends of RydC.

---

(**Figure 4B**). U24 and U47 contact N13 and likely make a hydrogen bonding interaction. Both U23/U24 and U46/U47 pairs are highly conserved in an alignment of the available RydC sequences (**Fröhlich et al., 2013**). The U24/U25 and U46/U47 interactions with the rim are consolidated by the N-terminal residues G4 and Q5 and likely include amide backbone and lysine side chain interactions (**Figure 4A,B**; distances provided in **Figure 4—figure supplement 1**).

Although highly conserved, the U23/U24 and U46/U47 pairs are not required for the base-pairing to the RydC target, *cfa* mRNA, because a chimeric RNA assembled from the RydC 5'-end (nt. 1 to 13) and the 3'-end of the sRNA MicA is as efficient in target gene regulation as the wild-type RydC (**Fröhlich et al., 2013**). To address the function of the highly conserved U residues, three mutant variants of RydC were constructed (**Figure 5A**). RydC-LI (U23G U24C), RydC-LII (U46C U47G), as well as RydC-LI/II (U23G U24C U46C U47G) were expressed from a constitutive promoter in wild-type *Salmonella* cells, and RNA stability was assessed by rifampicin treatment

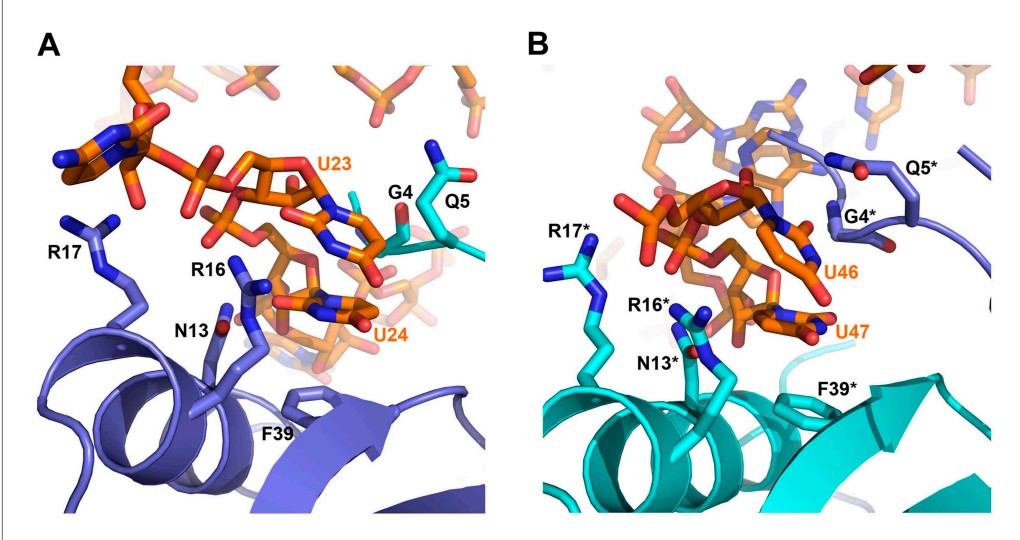

**Figure 4**. Contacts between the conserved U–U pairs of RydC and the Hfq rim. (**A**) The U23/U24 pair interacts with F39, R16, R17, and N13 on the convex rim of Hfq. The U23/U24 interactions with the rim are consolidated by the N-terminal residues G4 and Q5 of a neighbouring Hfq protomer (coloured in cyan). (**B**) The U46/U47 pair makes similar interactions with Hfq in a neighbouring asymmetric unit. Residues belonging to the symmetry related Hfq molecule are labelled with an asterisk (*) in panel **B**. Side chains are shown for all residues, main chain is shown for G4 in both panels to help visualize the residue position.

The following figure supplement is available for figure 4:

**Figure supplement 1**. Distances of the main contacts between Hfq and the U23/U24 and U46/U47 base steps of RydC.

(*Figure 5B*). Compared to wild-type RydC ($t_{1/2}$ ~32 min), mutant RydC-LII had a reduced half-life ($t_{1/2}$ ~11 min). Both variants RydC-LI and the double mutant RydC-LI/LII were even less stable ($t_{1/2}$ ~2 min) and especially the RydC-LI and RydC-LI/LII variants were less potent than the wild-type to regulate the reporter target Cfa–GFP (*Figure 5C*). The discrepancy between steady-state levels and regulatory potential observed between the different RydC variants could be the result of alternative binding patterns via LI and LII leading to changes in RNA orientation on Hfq. Thus, while functional conclusions may be limited at this point, our in vivo data suggest a requirement of the conserved U pairs in the loops of the RydC pseudoknot structure for RNA stability and indicate a functional importance for the interactions observed in the crystal structure of the complex.

The observed U23/U24 and U46/U47 interactions with the rim region may account for the findings from solution studies that *E. coli* Hfq has greater affinity for 16-mer polyU compared to the 6-mer $U_6$, because the longer polymer could form the contacts in the recessed core with the U-rich 3′-end and simultaneously interact with the rim through F39 and N13. The U23/U24 and U46/U47 interactions may also account for the partial fluorescence quenching observed for F39 in the presence of $U_6$ (*Robinson et al., 2013*), because excess $U_6$ may bind to the rim and interact with F39 in a similar way to the contacts observed for U23/U24 and U46/47 pairs of RydC.

Earlier studies identified residues R16, R17, R19, and K47 as forming a lateral RNA binding surface in *Salmonella* Hfq (*Sauer et al., 2012*; highlighted in cyan in *Figure 6C*). These residues are part of an arginine patch that is proposed to interact with both the sRNA and the mRNA molecules (*Sauer et al., 2012*). In our structure, R16 and R17 make phosphate backbone contacts near U23 and U24 (*Figure 4A*), while R19 and K47 are not interacting with the RNA. Residues R16 and R17 belonging to Hfq in the neighbouring crystal unit cell are also involved in the binding of the 5′-end of RydC (*Figure 3B*). The nucleotide stretch 8–12, representing the last portion of the RydC seed region, is in fact kept in an extended conformation by the interaction with helical region 8–17 from a symmetry related Hfq molecule (*Figure 3B*). This may account for observations from previous studies showing that mutations on this region of the rim do not impact on the ability

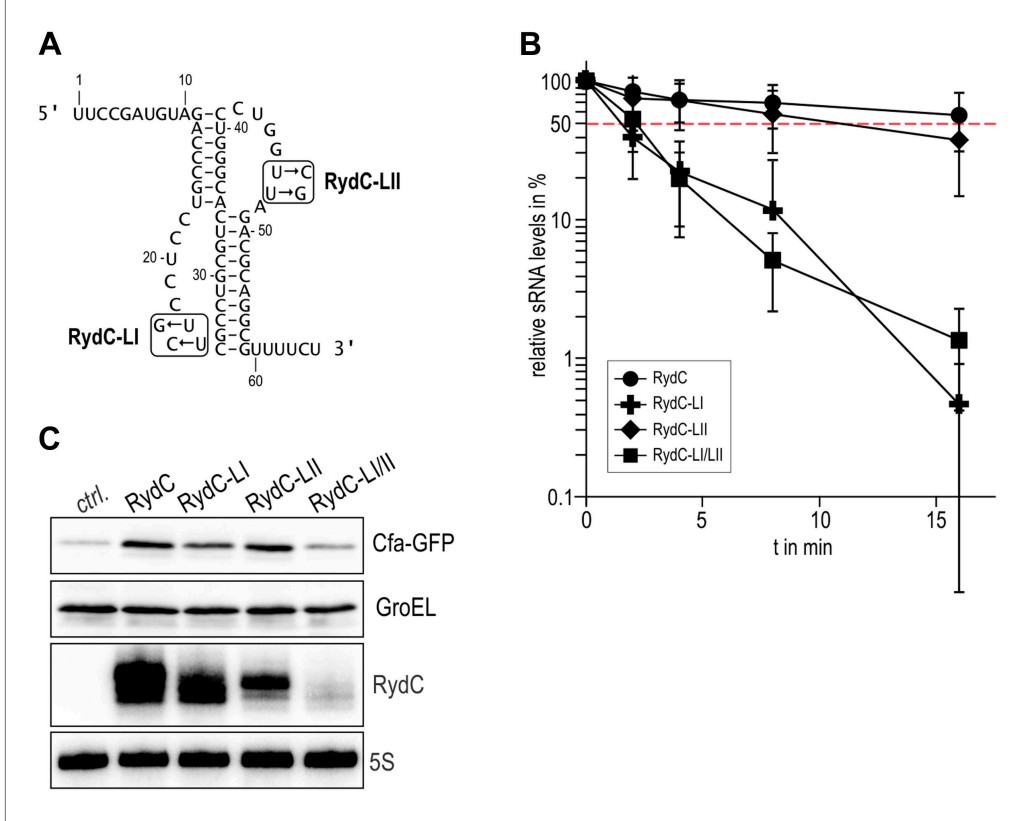

**Figure 5**. Requirement for the conserved UU steps for in vivo stability of RydC. (**A**) Schematic of RydC pseudoknot fold with substitutions of U23G and U24G (RydC-LI) and U46G and U47G (RydC-LII). (**B**) Stabilities of RydC and its variants RydC-LI, RydC-LII, and RydC-LI/LII. RNA levels were determined by Northern blot analyses. Total RNA samples were extracted prior to and at indicated time-points after inhibition of transcription by rifampicin in late exponential phase (OD$_{600}$ of 1) from *Salmonella* strain Δ*rydC* (JVS-0291) carrying plasmids expressing RydC (pKF42-1), RydC-LI (pKF224-9), RydC-LII (pKF223), or RydC-LI/II (pKF225) from the constitutive P$_L$ promoter. Error bars represent standard deviation calculated from three biological replicates. (**C**) Target regulation by RydC loop mutants. *Salmonella* Δ*rydC* (JVS-0291) express a plasmid-borne translational fusion of the RydC target *cfa* to the green fluorescent protein (Cfa–GFP; pKF31-1) in the presence of either a control (pJV300) or plasmids expressing RydC (pKF42-1), RydC-LI (pKF224-9), RydC-LII (pKF223), or RydC-LI/II (pKF225) from the constitutive P$_L$ promoter. Bacteria were grown to late exponential phase (OD$_{600}$ of 1) and Cfa–GFP and GroEL proteins were detected by Western blot (upper two panels). RydC variants as well as 5S RNA were detected by Northern blot (lower two panels).

of Hfq to bind sRNA sequences but rather on its capacity to stimulate annealing of sRNA with an mRNA target (*Panja et al., 2013*).

The RNA mediates numerous other interactions with neighbouring Hfq hexamers. One of the contacts mimics interactions of the A-A-N repeats on the distal face (*Figure 6—figure supplement 1*). A more extensive and potentially more interesting interaction occurs between the duplex region of RydC and the rim surface of an Hfq hexamer in a neighbouring asymmetric unit (*Figure 6A*). The residues mediating this interaction are R19, R66, P21, T49, Q33, and Q35 (*Figure 6B*).

To explore the contribution of the potential rim contacts to mediate the interaction between RydC and *cfa*, an Hfq mutant in the lateral rim (where residues R19, P21, Q33, Q35, T49, R66 have been mutated to A) was expressed and purified, and its binding rates and affinities were estimated from interferometry (ForteBio Octet Red96). For this experiment, the 5′ region of *cfa* (*cfa1*, nucleotides from TSS1 to −72 relative to the AUG, 139 nts, *Fröhlich et al., 2013*) was prepared with biotin attached to the 5′-end, which allowed the mRNA fragment to be immobilised on a streptavidin sensor surface. As shown in *Figure 6D*, both the wild-type protein (Hfq$^{WT}$) and the Hfq rim mutant (Hfq$^{Rim}$) have similar affinities and association rate constants for the immobilised *cfa*.

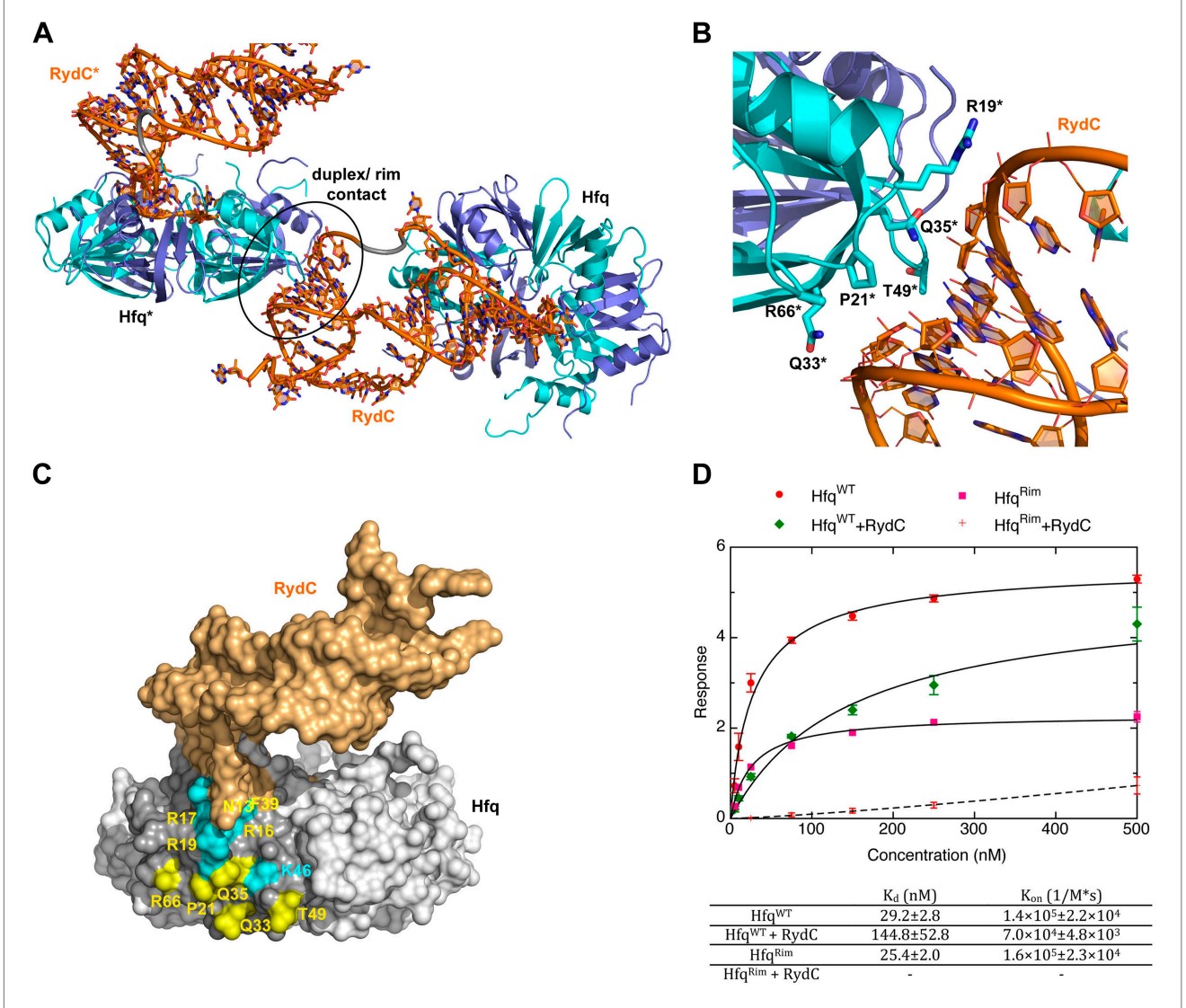

**Figure 6**. A potential RNA interaction surface on the circumferential rim of the Hfq hexamer. (**A**) Interaction between a duplex region of RydC and the rim surface of an Hfq molecule belonging to the neighbouring asymmetric unit in the crystal. Symmetry related RydC and Hfq are labelled with an asterisk (*). (**B**) A detailed view into the interaction site, showing the main protein residues making contacts with the duplex RNA. The side chain of R66 is not shown as it is disordered in the structure. (**C**) A surface representation of the RydC/Hfq complex, highlighting in colour in one protomer, the residues of the proposed extended rim that recognises RNA. The residues coloured cyan are from earlier studies that identified the rim as an RNA binding region (R16, R17, R19, K46; **Sauer et al., 2012**), and the yellow residues include an extended surface proposed here to engage duplex RNA (P21, Q33, Q35, T49, R66). Also labelled are the residues of this extended surface that engage the U–U steps in RydC (N13, R16, R17, F39; F39 and N13 are not visible in this view). (**D**) Rates and equilibrium constants for binding of Hfq and Hfq/RydC to *cfa* for wild-type (HfqWT) and rim-mutant protein (HfqRim; R19A, P21A, Q33A, Q35A, T49A, R66A). The measurements were performed by fitting the averaged responses from three independent experiments. It was not possible to fit the weaker binding response of HfqRim binding RydC.

The following figure supplements are available for figure 6:

**Figure supplement 1**. Interactions between RydC and the proximal face of a symmetric Hfq molecule.

**Figure supplement 2**. Binding rates and equilibrium constants for Hfq[66] and Hfq[66]/RydC to immobilized *cfa*.

In the presence of RydC, the association of Hfq with *cfa* changes. The binding affinity of Hfq[WT] for *cfa* decreases by a factor of roughly 5, suggesting that the formation of the Hfq–sRNA complex affects the Hfq affinity for the mRNA fragment. Moreover, the presence of RydC decreases

by an order of magnitude the association rate constant of Hfq for *cfa*. This could result from *cfa* being bound in a slightly different way in the Hfq/*cfa* complex as compared to the Hfq/RydC/*cfa* ternary complex. Strikingly, in the presence of RydC, the Hfq[Rim] mutant no longer has detectable binding to *cfa*. This finding suggests that the RydC interacts differently with Hfq[WT] and the Hfq[Rim] lateral surface mutant in a way that perturbs *cfa* binding. Thus, the rim interactions shown in *Figure 6A,B,C* are crucial for forming a productive sRNA:Hfq:mRNA assembly.

### The N- and C-termini of Hfq interact distributively with RydC

The C-terminal regions of Hfq, beyond residue 70, are generally disordered, but density could be observed for some of the protomers into which the model could be partially extended. Although poorly ordered, the C-terminal tails appear to be making distributive contacts over the surface of RydC. The N-terminal regions of three Hfq protomers, exposed on the proximal face of the protein, also interact with RydC, mainly in the vicinity of nucleotides 23–25, 51–52, and 59.

To explore if the observed interactions between the Hfq C-termini and RydC might be important in vivo, we compared the stabilities of the sRNA in the presence of wild-type Hfq and an Hfq mutant truncated at residue 70 (*Figure 7*). We observed that the half-life of RydC was greatly reduced in the absence of the C-terminal end ($t_{1/2}$ ~5 min) and was comparable with the half-life observed for an *hfq* null mutant ($t_{1/2}$ ~4 min). However, upon initial decay of a subfraction of RydC, the remaining population appears to be more stable and is only degraded at a rate comparable to RNA in the wild-type background. In vitro binding experiments show that the affinity of purified Hfq truncated at residue 66 (Hfq[66]) for binding *cfa* is similar to that of the wild-type protein; however, the presence of RydC decreases substantially the binding of Hfq[66] (*Figure 6—figure supplement 2*). The curve for Hfq[66]/RydC binding to *cfa* could not be modelled, but the affinity is likely to be reduced by two orders of magnitude compared with the binding of Hfq[66] alone to *cfa.* This reduction is much greater than the fivefold decrease in affinity when comparing the mRNA binding of the wild-type Hfq and its Hfq/RydC complex (*Figure 6D*). These observations suggest that the C-terminal tails are important for facilitating the formation of the sRNA:mRNA:Hfq ternary complex.

The C-terminal tails appear to influence the rate of Hfq–RNA complex formation, as indicated by the order-of-magnitude lower on-rate of Hfq[66] compared to Hfq[WT] for binding *cfa* (*Figure 6D* and *Figure 6—figure supplement 2*). Moreover, the interactions of the tails with sRNA may explain the puzzling observation that the Hfq[WT]–RydC has a lower $k_{on}$ than Hfq[WT] for association with *cfa*. In this case, we envisage that the tails, sequestered with the sRNA, are no longer available to 'fish' for and accelerate association with the mRNA partner.

## Discussion

The crystal structure of RydC/Hfq suggests key elements of molecular recognition of this effector complex. We observe interactions of the 3′ polyU tract with the recessed pore in the Hfq proximal face and interactions of U–U dinucleotide steps in single-stranded regions with the surface near the rim. Based on results from in vivo crosslinking and RNA sequencing from enterohemorrhagic *E. coli*, it has been proposed that the consensus Hfq binding site on many sRNAs includes a U–U dinucleotide associated with an unpaired region (*Tree et al., 2014*). Our results are consistent with this prediction and provide a structural rationalisation: the unpaired region is required to engage the U–U step on the rim. Interactions with the rim may account for the capacity of longer U-tracts to bind Hfq more strongly (*Murina et al., 2013*; *Panja et al., 2013*), since these are anticipated to bridge between the rim and the 3′-polyU-binding pore. The rim might

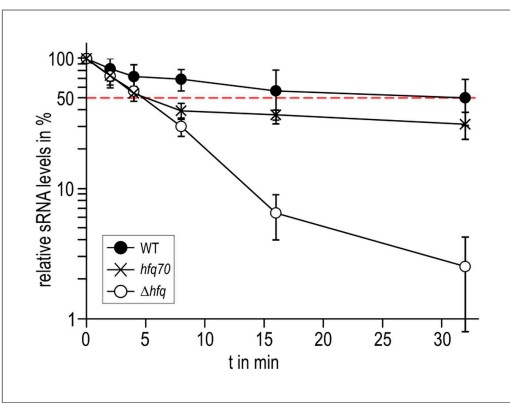

**Figure 7**. Role of the Hfq C-terminus in RydC stability in vivo. Stabilities of RydC were determined by Northern blot analyses in *Salmonella* strains Δ*rydC* (WT; JVS-0291), Δ*rydC* Δ*hfq* (Δ*hfq*; JVS-10665), and Δ*rydC hfq*70 (*hfq*70; JVS-11150) expressing RydC from the constitutive P$_L$ promoter (pKF42-1). Total RNA samples were extracted prior to and at indicated time-points after inhibition of transcription by rifampicin in late exponential phase (OD$_{600}$ of 1). Error bars represent standard deviations calculated from three biological replicates.

also be the binding site for the internal U-rich region in SgrS that is required for stable association with Hfq (*Ishikawa et al., 2012*).

The crystal structure suggests that one RydC can be sandwiched between two Hfq hexamers, and a similar model has been proposed for interaction of Hfq with the sRNA DsrA (*Wang et al., 2011*). However, solution data indicate that the stoichiometry of the complex is sensitive to buffer conditions and that a 1:1 complex is preferred at higher salt concentrations, but a 2:1 Hfq hexamer:RydC complex can be formed in lower ionic strength buffer (*Figure 2—figure supplement 2*). We propose that the 1:1 complex is the physiologically relevant species in vivo and that the sandwiching complex we observe in the crystal is likely to mimic a folding intermediate en route to forming a closed 1:1 RydC:Hfq complex. Accordingly, the contacts formed by the RydC with the neighbouring Hfq hexamers can both occur on the proximal face of the same, single hexamer to form the 1:1 complex (shown schematically in *Figure 8*).

Solution studies indicate that the C-termini of *Vibrio cholerae* and *E. coli* Hfq are natively unstructured (*Beich-Frandsen et al., 2011*; *Vincent et al., 2012*). For most of the Hfq subunits in the crystal structure studied here, the C-terminal regions beyond residue 77 are disordered, but broken density in the vicinity is likely to be due to multiple conformations of the C-terminal region. These regions make consolidating interactions with the RNA and appear to be distributed over the exposed surfaces of the nucleic acid. A structural condensation of the C-terminal domains with sRNA binding is also consistent with data from X-ray and neutron solution scattering (*Henderson et al., 2013*). Both the structural data as well as in vivo experiments suggest that the C-terminal end of Hfq contributes to the

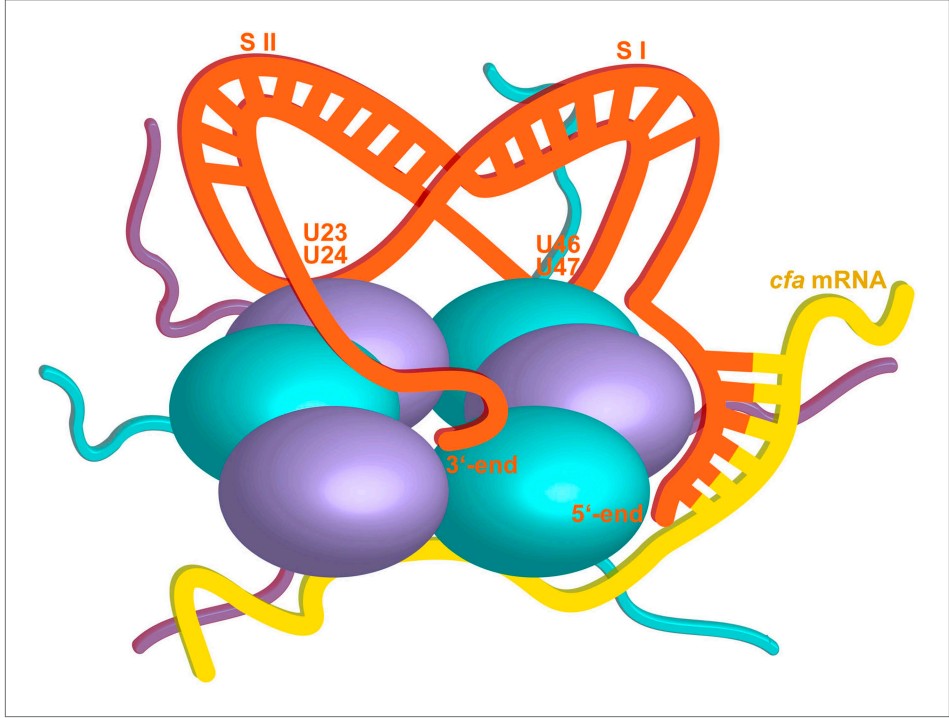

**Figure 8**. Hypothetical model of the RydC/mRNA/Hfq effector complex. Portions of the protomers forming the Hfq core have been represented with six spheres, from which the disordered C-terminal tails extend radially. RydC (orange) sits on the proximal face of Hfq, with the 3'-end U-rich tail interacting with the central channel and the two conserved U–U pairs (U23/U24 and U46/U47) making contacts with the lateral face of the hexamer. The two double strands conferring the pseudoknot structure to RydC are indicated as S I and S II. The target mRNA *cfa*, depicted in yellow, associates with the distal face of Hfq, and it is proposed to form a duplex with the 5'-end 'seed' region of RydC that is recognised by the circumferential rim of Hfq. The six long C-termini are depicted in the cartoon, and the shorter N-termini are not shown for clarity. The association of the two RNA partners is aided by the C-terminal tails of Hfq, which extend towards the RNA molecules, embracing them and stabilizing the seed/target pairing. The natively unstructured termini make distributed interactions with the RNA, visiting dynamically multiple points on the nucleic acid.

association with RydC. The distributive interactions of the C-terminal domain are not expected to provide specificity to the recognition but may enhance the stability of equilibrium complexes as well as boosting the association rates by increasing the effective molecular encounter radius. The distributive interaction may account for the finding that deletion of the C-terminal extension decreases binding to mRNAs (*Večerek et al., 2008*) and that the C-terminal tail can bind to some RNA species (*Robinson et al., 2013*). The same might be true for yeast Lsm1 mutants with C-terminal truncations that have reduced affinity for RNA (*Chowdhury et al., 2012*).

These results are in agreement with a study that compared the distribution of *Salmonella* RNAs associating with Hfq variants of different species (*Sittka et al, 2009*). Similar to several orthologues of various bacterial species, Hfq of the archaeon *Methanocaldococcus jannaschii* lacks the unstructured C-terminus. *M. jannaschii Hfq* is generally able to complement a *Salmonella hfq* null mutant; however, the distribution of associated sRNAs recovered from pull-down experiments varies greatly compared to the wild-type. While a number of *Salmonella* sRNAs (including, for example, the Hfq-dependent SroC, InvR, and GcvB) are not reduced in their cellular abundance, a second class of sRNAs is strongly affected by the absence of the C-terminal end. This latter group includes RydC and also several sRNAs highly abundant in the wild-type (e.g., RprA, SdsR, and ArcZ), and it is tempting to speculate that this subpopulation of sRNAs is associated with Hfq through supporting interactions with the C-terminal tail. The differing roles of the tails for different sRNAs may account for the apparently conflicting results reported in the literature on the effects of the C-terminus on sRNA binding and activity (*Olsen et al., 2010*). Finally, it should be noted that the C-terminal tails have different lengths in different Hfq homologs, and it is likely that their role differs in different species (*Vincent et al., 2012*).

As a general principle, the interactions of natively unstructured regions with RNA may be important in other nucleic acid/protein complexes (*Tompa and Csermely, 2004*; *Kucera et al., 2011*; *Jonas and Izaurralde, 2013*), and they can contribute to cooperative binding (*Hunter and Anderson, 2009*; *Motlagh et al., 2014*). We envisage that these interactions would help to keep the sRNA and mRNA in proximity without constraining them, so that the mRNA might travel along the distal face while the sRNA was held in a more fixed conformation on the proximal face. In this way, the complex can glide along an mRNA until the seed and target region meet. Once this occurs, they will close like a zipper to form a short duplex that could be engaged on the convex rim of the Hfq. The mismatch of molecular symmetries and the high flexibility of both the N- and C-termini of Hfq ensure that the complex will be intrinsically heterogeneous in conformation, but highly dynamic.

Based on the current crystal structure, including the lattice interactions observed, and on other available structural and biophysical data, we propose a speculative model for the ternary complex of Hfq/sRNA and mRNA target (*Figure 8*). The model shares some similarity to the schematic proposed by *Panja et al. (2013)* and predicts that the 5′ seed will be exposed and available for interaction with the target to form a duplex region that binds to the convex rim of the Hfq hexamer. We envisage that a different model will be required to explain the interactions of sRNA using recognition regions that are internal and not in the 5′ seed. One prediction is that the base-pairing sites in this group of sRNAs will be presented near the rim of Hfq. This model awaits testing.

## Materials and methods

### Expression and purification of Hfq

Wild-type *E. coli* Hfq was expressed and purified as previously described by *Bandyra et al. (2012)*. Hfq mutants were expressed from pBAD vector in Top10 Δ*hfq* strain (JVS-02001) to avoid hetero-hexamer formation. The mutant proteins were purified with the same protocol used for the wild-type Hfq, with a HiTrap Heparin purification step added (buffer A: 50 mM Tris, pH 8.0, 100 mM NaCl, 100 mM KCl; buffer B: buffer A + 1 M NaCl). The presence of mutations or truncations was confirmed by mass spectrometry analyses.

### Preparation of RydC RNA

For *Salmonella* RydC RNA in vitro transcription (IVT), a DNA template was amplified by PCR from pKF42-1 using primers RydC_T7_fwd and RydC_T7_rev (*Table 1*).

Transcription mixtures containing DNA-template, ribonucleotides (rNTPs), T7 RNA polymerase, DTT (dithiothreitol), and RNase Out (Invitrogen, UK) were incubated in transcription buffer (400 mM Tris (pH 7.9/8.0), 250 mM MgCl$_2$, 20 mM spermidine) for 4 hr at 37°C, and then Turbo DNase (Ambion)

was added to digest the template. The IVT product was separated on a 10% polyacrylamide gel containing 7 M urea. The band corresponding to the transcribed RydC was visualized by UV-shadowing, excised from the gel, and the RNA was electro-eluted overnight using an EluTrap system (Whatman, UK). The RNA sample was concentrated by ultrafiltration through a Vivaspin 500 concentrator (5 kDa cut-off).

## Crystallization of Hfq–RydC complex

The RNA–protein complex was prepared by mixing purified $Hfq_6$ and RydC in a ratio of 2:1, the Hfq hexamer concentration being 15 mg/ml. Crystals of the Hfq–RydC complex were grown by sitting drop vapour diffusion after adding an equal volume of crystallization buffer (0.2 M trisodium citrate, 0.1 M sodium cacodylate (pH 6.5), 15% vol/vol isopropanol) to the protein–RNA mixture. Crystals were harvested using 25% PEG400 as cryoprotectant and flash frozen in liquid nitrogen. X-ray data were collected on station I24 at Diamond Light source. Data were collected at 100 K at a wavelength of 0.9778 Å.

Images obtained from the two best crystals having the same space group and cell dimensions were merged and scaled together, before proceeding with the structure determination.

## Crystal structure determination

The structure was solved by molecular replacement with PHASER (*McCoy et al., 2007*) using residues 5–66 of *E. coli* Hfq as the search model. Electron density for RNA was apparent in the early maps. Maps were improved with density modification with PARROT using histogram matching with tRNA/aminoacyl-tRNA synthase experimental structure factors as the reference distribution. This structure has a similar solvent content and RNA/protein mass ratio as the RydC/Hfq crystals, and density modification with this method improved the map and visualisation of the RNA. The structure was refined using BUSTER (*Bricogne et al., 2011*) and with jelly body restraints in refinement with REFMAC (*Murshudov et al., 2011*). The model was built using COOT (*Emsley et al., 2010*). The protein and RNA stereochemistry were validated by using both Coot validation tools and Procheck from the CCP4 suite (*Laskowski et al., 1993*; *Emsley et al., 2010*). The Ramachandran plot of the model places 92.9% of residues in most favoured regions, 5.4% in additional allowed regions, 1.6% in generously allowed regions, and none in disallowed regions. X-ray data collection and refinement statistics are summarized in *Table 2*. Figures were prepared using PYMOL (*DeLano, 2006*). The coordinates and structure factors have been deposited in the PDB with accession code 4v2s.

**Table 1.** Oligonucleotides

| Name | Sequence 5′-3′ |
| --- | --- |
| RydC_T7_fwd | GTTTTTTTTTTAATACGACTCACTATAGGCTTCCGATGTAGACCCGT |
| RydC_T7_rev | CAGAAAACGCCTGCGTCTAACCAGGACCCG |
| JVO-0322 | CTACGGCGTTTCACTTCTGAGTTC |
| JVO-4363 | AGAAAACGCCTGCGTC |
| JVO-4558 | GTTTTTTTTTAATACGACTCACTATAGGTTGTTTATATTACGATAATT |
| JVO-4721 | GTTTTTTTTTTAATACGACTCACTATAGGTTCCGATGTAGACCCGTCC |
| JVO-4722 | AGAAAACGCCTGCGTCTAAC |
| JVO-5165 | GTTTTTTTTTTAATACGACTCACTATAGGTTCCGATGTAGAGCGGTCC |
| JVO-9044 | CCCACGGACAATTCCGT |
| JVO-10909 | CAGCAACAATGCCGGTGGCGGCGCCAGCAATAACTACCATTAAGGTCCATATGAATATCCTCCTTAG |
| JVO-10910 | ATTATCCGACGCCCCCGACATGGATAAACAGCGCGTGAACGTGTAGGCTGGAGCTGCTTC |
| JVO-10913 | CGAGACGCAGGCGTTTTC |
| JVO-10914 | P-CCAGGACCCGTGACG |
| JVO-10915 | GCCGCCTGCGTCACGG |
| JVO-10916 | P-GGAGGACGGGTCTACATC |
| | P: 5′ phosphorylation |

**Table 2.** X-ray data collection and refinement statistics

| Data collection | |
|---|---|
| Space group | $P2_12_12_1$ |
| Cell dimensions | |
| $a$, $b$, $c$ (Å) | 71.94, 73.36, 137.95 |
| $\alpha$, $\beta$, $\gamma$ (°) | 90, 90, 90 |
| Resolution (Å) | 24.47–3.48 (3.69–3.48)* |
| $R_{merge}$ | 0.152 (0.886) |
| $I/\sigma I$ | 9.6 (2.3) |
| Completeness (%) | 99.1 (99.4) |
| Multiplicity | 5.8 (5.9) |
| Refinement | |
| Resolution (Å) | 24.32 (3.48) |
| No. reflections | 9190 |
| $R_{work}/R_{free}$ | 0.22/0.28 |
| No. atoms | 4470 |
| Protein | 3238 |
| RNA | 1187 |
| Water | 45 |
| $B$-factors | |
| Overall | 94.69 |
| Protein | 93.36 |
| RNA | 122.60 |
| Water | 54.90 |
| R.m.s. deviations | |
| Bond lengths (Å) | 0.010 |
| Bond angles (°) | 1.475 |

*Values in parentheses are for the highest-resolution shell.

## Binding rates and affinity measurements

Kinetic measurements with Bio-Layer Interferometry were performed using an Octet RED96 equipped with Streptavidin sensors (ForteBio, UK) on 96-well plates. The experiment was performed in the binding buffer (25 mM Tris, pH 7.5, 50 mM NaCl, 50 mM KCl, 1 mM $MgCl_2$, 1 mM DTT), which was also used to prepare all dilutions, for dissociation and neutralisation. A fragment of *cfa* mRNA (*cfa1*, nucleotides from TSS1 to −72 relative to the AUG, 139 nts, *Fröhlich et al., 2013*) was in vitro transcribed from a PCR template (JVO-4558*JVO-9044 on pKF31-1) and labelled with biotin in an IVT reaction using fivefold excess of GMP-biotin (TriLink Biotechnologies, San Diego CA, USA) over GTP. 5′ biotin labelled *cfa* fragment was immobilised on the biosensor that was subsequently submerged into 10 µM solution of maltose binding protein (MBP) labelled with biotin. The binding of wild-type and mutant Hfq were assayed at 0, 5, 10, 25, 75, 150, 250, and 500 nM protein over 400 s, in the absence or presence of 1 µM chemically synthesized RydC (Dharmacon, GE Healthcare, UK). The dissociation was monitored over 300 s and was followed by regeneration of the sensors using 1 M $MgCl_2$. Another set of tips was saturated with MBP-biotin and the measurements were then repeated for all Hfq and Hfq–RNA concentration series. The data were fitted with Data Analysis software (ForteBio) with a 1:1 binding model. The plots were prepared with Profit (Quantum Soft, Switzerland) using the following equation for response fit:

$$Y = Rm \times X^n/(K_d^n + X^n),$$

where Y is the observed binding, X is the molar concentration of the ligand, Rm is the maximum specific binding, and n is the Hill's coefficient.

## DNA/RNA oligonucleotides and plasmids

Sequences of all oligonucleotides employed in this study are listed in *Table 1*. All plasmids used in this study are summarized in *Table 3*.

For plasmids expressing loop mutants of RydC from the $P_L$-promoter, plasmid pKF42-1 served as template for PCR amplification with primer pairs JVO-10915/JVO-10916 (pP$_L$-RydC-LI; pKF224-9), JVO-10913/JVO-10914 (pP$_L$-RydC-LII; pKF223), and the linear fragments were purified and self-ligated. Similarly, pP$_L$-RydC-LI/LII (pKF225) was constructed by self-ligation of a PCR product of JVO-10915/10,916 using pKF223 as template. Competent *E. coli* TOP10 were used for all cloning purposes.

## Bacterial strains and growth conditions

For the in vivo analyses, bacteria were grown aerobically in L-Broth (LB) medium at 37°C. Where appropriate, liquid and solid media were supplemented with antibiotics at the following concentrations: 100 µg/ml ampicillin, 50 µg/ml kanamycin, and 20 µg/ml chloramphenicol.

A complete list of bacterial strains employed in this study is provided in *Table 4*. *Salmonella enterica* serovar Typhimurium strain SL1344 (JVS-0007) is referred to as wild-type strain and was

**Table 3.** Plasmids

| Plasmid trivial name | Plasmid stock name | Relevant fragment | Comment | Origin, marker | Reference | Used in Figure |
|---|---|---|---|---|---|---|
| ctrl. | pJV300 | | pP$_L$ control plasmid, expresses an ~50 nt nonsense transcript derived from *rrnB* terminator | ColE1, Amp$^R$ | (Sittka et al., 2007) | 5C |
| pP$_L$-RydC | pKF42-1 | RydC | Expresses *Salmonella* RydC from constitutive P$_{LlacO}$ promoter | ColE1, Amp$^R$ | (Fröhlich et al., 2013) | 1B/C; 5B/C; 7 |
| pP$_L$-RydC-SI | pKF60-1 | RydC-SI | Expresses *Salmonella* RydC-SI (SNEs G37C; G39C) from constitutive P$_{LlacO}$ promoter | ColE1, Amp$^R$ | (Fröhlich et al., 2013) | 1B/C |
| pP$_L$-RydC-LI | pKF224-9 | RydC-LI | Expresses *Salmonella* RydC-LI (SNEs U23G; U24C) from constitutive P$_{LlacO}$ promoter | ColE1, Amp$^R$ | This study | 5B/C |
| pP$_L$-RydC-LII | pKF223 | RydC-LII | Expresses *Salmonella* RydC-LII (SNEs U46C; U47G) from constitutive P$_{LlacO}$ promoter | ColE1, Amp$^R$ | This study | 5B/C |
| pP$_L$-RydC-LI/LII | pKF225 | RydC-LI/LII | expresses *Salmonella* RydC-LI/LII (SNEs U23G; U24C and U46C; U47G) from constitutive P$_{LlacO}$ promoter | ColE1, Amp$^R$ | This study | 5B/C |
| *cfa::gfp* | pKF31-1 | *cfa::gfp* | expresses *cfa::gfp* translational fusion (−210 rel. to AUG + 15 codons of *cfa*) from constitutive P$_{LtetO-1}$ promoter | pSC101*, Cm$^R$ | (Fröhlich et al., 2013) | 5C |
| | pKD4 | | Template plasmid for Km$^R$ mutant construction | oriRγ, Amp$^R$ | (Datsenko and Wanner, 2000) | |
| | pKD46 | *γ-β-exo* | temperature-sensitive plasmid to express λRED-recombinase from arabinose-inducible P$_{araB}$ promoter | oriR101, AmpR | (Datsenko and Wanner, 2000) | |
| | pCP20 | FLP–ci857 | Temperature-sensitive Flp recombinase expression plasmid | pSC101, Amp$^R$, Cam$^R$ | (Cherepanov and Wackernagel, 1995) | |
| Hfq wild-type | pEH–10–(hfq) | Hfq | Expresses wild-type Hfq protein without tags | Amp$^R$ | Kind gift of I. Moll (Max F. Perutz Laboratories, University of Vienna, Austria) | 2, 3, 4, 6 |
| Hfq$^{Rim}$ | pBAD-Rim | Hfq R19A,P21A,Q33A, Q35A,T49A,R66A | Expresses mutant of Hfq protein without tags | pMB1, Amp$^R$ | This study | 6 |
| Hfq$^{66}$ | pBAD-66 | Hfq amino aids 1-66 | Expresses truncated Hfq protein without tags | pMB1, Amp$^R$ | This study | S5 |

**Table 4.** Strains

| Trivial name | Stock | Genotype; relevant markers | Source/reference | Used in Figure |
|---|---|---|---|---|
| *Salmonella* | | | | |
| | JVS-0007 | SL1344; Str$^R$ *hisG rpsL xyl* | Laboratory stock | |
| Δ*rydC* | JVS-0291 | Δ*rydC*::Kan$^R$ | (*Papenfort et al., 2008*) | 1B/C; 5B/C; 7 |
| Δ*hfq* | JVS-0584 | Δ*hfq* | (*Sittka et al., 2007*) | 1B/C |
| Δ*rydC* Δ*hfq* | JVS-10665 | Δ*rydC*::Kan$^R$ Δ*hfq* | (*Fröhlich et al., 2013*) | 7 |
| | JVS-10701 | *hfq70*::STOP | This study | |
| Δ*rydC hfq70* | JVS-11150 | *hfq70*::STOP Δ*rydC*::Kan$^R$ | This study | 7 |
| *E. coli* | | | | |
| TOP10 | JVS-2000 | F⁻ *mcrA* Δ(*mrr-hsdRMS-mcrBC*) Φ80*lacZ*Δ*M15* Δ*lacX74 recA1 araD139* Δ(*ara-leu*)7697 *galU galK rpsL endA1 nupG* λ⁻ | Invitrogen | |

used for mutant construction. Single mutant derivatives were constructed by the λRed recombinase one-step inactivation method. To obtain JVS-10701 (SL1344 *hfq70*::STOP), *Salmonella* cells carrying the pKD46 helper plasmid were transformed with a DNA fragment amplified from pKD4 using JVO-10909/JVO-10910. Subsequently, the Kan$^R$ cassette of λRed-derived mutants was eliminated by transformation with the FLP recombinase expression plasmid pCP20 (*Datsenko and Wanner, 2000*). Phage P22 transduction was used to transfer chromosomal modifications to a fresh *Salmonella* wild-type background as well as to obtain strains carrying multiple mutations.

## Protein sample analysis

To prepare whole-cell samples for Western blot analyses, bacteria were collected by centrifugation (16,000×*g*; 2 min; 4°C), and pellets were resuspended in 1X protein loading buffer (Fermentas, UK) to a final concentration of 0.01 OD/µl. Samples corresponding to 0.1 OD were loaded per lane and resolved by SDS-PAGE, after which proteins were transferred to PVDF membranes as described in *Sittka et al. (2007)*. GFP fusion proteins and GroEL were detected using commercially available antibodies (GFP: 1:5000; mouse; Roche # 11814460001 and GroEL: 1:10,000; rabbit; Sigma-Aldrich G6532). Anti-mouse or anti-rabbit secondary antibodies conjugated with horseradish peroxidase (1:10,000; GE Healthcare) were used in all cases. Signals were visualized using the Western Lightning reagent (PerkinElmer, Waltham MA, USA) and an ImageQuant LAS 4000 CCD camera (GE Healthcare).

## Determination of RNA half-life

To monitor RNA half-life of RydC mutant variants, cells were grown to an OD$_{600}$ of 1 and treated with rifampicin (final concentration: 500 µg/ml) to abrogate transcription. RNA samples were withdrawn at indicated time-points, and RNA decay was determined by Northern blot analysis as previously described (*Urban and Vogel, 2007*; *Fröhlich et al., 2012*). RydC and its variants were detected with the universal oligo JVO-4364; 5S RNA served as loading control (JVO-0322).

## Electrophoretic mobility shift assay (EMSA)

Formation of complexes between sRNAs and Hfq in vitro was analysed by gel shift assays. For RNA in vitro synthesis, ~200 ng of template DNA carrying a T7 promoter sequence was amplified by PCR (RydC: JVO-4721*JVO-4722 on pKF42-1; RydC-S1: JVO-5165*JVO-4722 on pKF60-1) and reverse transcribed and 5′-end-labelled as described previously (*Fröhlich et al., 2013*). Labelled RNA RydC or RydC-S1 (4 pmol) was denatured (95°C, 2 min), chilled on ice for 5 min, and supplemented with 1× structure buffer and 1 µg yeast RNA. Upon addition of purified Hfq (concentration as indicated in the figure legends) or Hfq dilution buffer (control; 1× structure buffer, 1% (vol/vol) glycerol, 0.1% Triton X-100), samples were incubated at 37°C for 10 min. Unlabelled competitor RNA was added at the indicated concentrations, and samples were incubated for additional 10 min. Prior to loading, reactions were mixed with native loading buffer (50% glycerol, 0.5× TBE, 0.02% (wt/vol)

bromophenol blue) and separated by native PAGE (6% PAA). Gels were dried and signals were determined on a Typhoon FLA 7000 phosphorimager.

## Acknowledgements

This work was supported by the Wellcome Trust (BFL) and the Bavarian BioSysNet programme (JV). We thank the beamline staff at Diamond Light Source for help and advice with data collection. We thank Len Packman for mass spectrometry analysis of proteins and Barbara Plaschke for excellent technical assistance. We thank Tommaso Moschetti and Katherine Stott for advice for the binding experiments, Kiyoshi Nagai for helpful comments, Steven Hardwick for help and advice with the AUC, and Tristan Croll for analysis of the stereochemistry of the refined model.

## Additional information

### Funding

| Funder | Grant reference number | Author |
| --- | --- | --- |
| Wellcome Trust | RG61065 | Ben F Luisi |

The funder had no role in study design, data collection and interpretation, or the decision to submit the work for publication.

### Author contributions

DD, Conception and design, Acquisition of data, Analysis and interpretation of data, Drafting or revising the article, Contributed unpublished essential data or reagents; KSF, KJB, BFL, Conception and design, Acquisition of data, Analysis and interpretation of data, Drafting or revising the article; HAB, Acquisition of data, Analysis and interpretation of data, Drafting or revising the article; SH, Acquisition of data, Analysis and interpretation of data; JV, Conception and design, Analysis and interpretation of data, Drafting or revising the article

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
