## [Decision Letter]

Thank you for sending your work entitled “Recognition of the small regulatory RNA RydC by the bacterial Hfq protein” for consideration at *eLife*. Your article has been favourably evaluated by John Kuriyan (Senior editor) and 3 reviewers, one of whom is a member of our Board of Reviewing Editors.

The Reviewing editor and the other reviewers discussed their comments before we reached this decision. All three thought the structure of a full small regulatory RNA bound to Hfq was a major advance. Their comments listed below are related to the need for clarification of some data.

1) Figure 1: Is it clear that the wild-type RydC and the RydC-S1 mutant, while binding Hfq with similar affinities (cited as results not shown), bind to the same face and in the same way to Hfq? One question is how much the findings with this pseudoknot RNA can be extrapolated to other Hfq-binding sRNAs. Figure 1 shows that RydC-S1 expressed in cells in the presence of Hfq is at a very low level at the time rifampicin is added (t=0). Hfq may bind to RydC-S1, but it apparently is unable to protect it from degradation. The data clearly shows the presence of Hfq inhibits degradation of RydC. This could be an indirect effect, but it could be due to Hfq binding/stabilizing the RydC pseudoknot fold and inhibiting degradation from the 5' end. The conclusions and comments here should be clarified.

2) Figure 2: The major issue from the crystallography is that in the crystal, RydC is interacting with two adjacent Hfq hexamers. The authors suggest that the contacts seen on neighboring hexamers can also form on a single one, and that such single interaction is likely to be the functional complex. A clearer discussion of what interacts with each hexamer would at least clarify whether there are mutational tests that might demonstrate the importance of each type of interaction. Related to this point, Figure 2 is the only figure that provides a picture of the whole structure and therefore is important for readers to understand the major conclusions, as well as the nature of the complex with two Hfq hexamers. It might be easier to understand this if the second RydC (at upper right) in 2b were in a different shade, for instance (lighter), and possibly if a schematic were included to make it clear what is going on (what the contacts are on each proximal face). Highlighting in some fashion (maybe box and expand to the side) the rim interaction region is important in trying to understand the implications of this sandwiched structure for the rim interaction. It is hard to read the label in some cases. Is the U-rich tail on one hexamer and the 5' end on the other Hfq hexamer? How important is the 5' end for function?

3) Figure 5: The data in this figure is difficult to understand. By stability measurements alone, LI and LI/LII should be similarly defective; however, for activity, LI is much more active than LI/LII, and in terms of accumulation of RydC, LI looks better than LII (or the double). Is there an explanation for this?

4) Figure 6: Why was an Hfq mutant with 6 changes used for this experiment? It is difficult to know which rim interactions are in fact important, given this drastic change. Is this active for RydC regulation of *cfa* in vivo? Is this mutant derivative present at wild type levels in the cell? Does the lateral rim mutant show wild type binding to *cfa* in the absence of RydC and reduced binding in the presence of RydC in vivo?

Also, no data is shown on response vs time to obtain the kinetic rates for the SPR experiments in Figure 6. This data and the fit should be added to supplementary information. The response values shown in Figure 6 are low and raise a concern on the accuracy of the evaluated parameters. Additionally, the asymptotic maximum response value for the Hfqrim (1.8) is less than half than that for HfqWT (5.0). If both Hfq's bind *cfa* RNA in a similar way what accounts for this difference? The authors could support their conclusions from the SPR data by including electrophoretic gel shift data on HfqWT and Hfqrim binding to the *cfa* RNA and RydC.

5) Figure 7: The authors need to comment on these decay curves. While they emphasize here the very early points (half-life reduced to 5', compared to hfq null), it is striking that only a fraction of the RNA is in fact degraded (40%?) in this mutant. Does that suggest that a subpopulation becomes sensitive, but an almost equal part is stably bound to Hfq? Is this truncated Hfq active for RydC regulation in vivo?

6) Discussion: The authors should comment on the fact that some Hfq homologs do not have this extension as well as name the proteins that have unstructured regions important to other nucleic acid/protein complexes.

---

## [Author Response]

*1)*
Figure 1*: Is it clear that the wild-type RydC and the RydC-S1 mutant, while binding Hfq with similar affinities (cited as results not shown), bind to the same face and in the same way to Hfq? One question is how much the findings with this pseudoknot RNA can be extrapolated to other Hfq-binding sRNAs.*
Figure 1
*shows that RydC-S1 expressed in cells in the presence of Hfq is at a very low level at the time rifampicin is added (t=0). Hfq may bind to RydC-S1, but it apparently is unable to protect it from degradation. The data clearly shows the presence of Hfq inhibits degradation of RydC. This could be an indirect effect, but it could be due to Hfq binding/stabilizing the RydC pseudoknot fold and inhibiting degradation from the 5' end. The conclusions and comments here should be clarified*.

To address the question of whether RydC and RydC-S1 bind the same face of Hfq, we performed EMSA competition experiments. Our results demonstrate that the affinity for Hfq of the S1 mutant is lower than that of wild type RydC (Figure 9 panel A). The complex formed by RydC with Hfq can only be partially outcompeted in the presence of a 500-fold excess of cold RydC-S1 (Figure 9 panel B, lanes 1-7). In contrast, RydC-S1/Hfq complexes appeared more susceptible to competition and were already destabilized in the presence of 10-fold excess of unlabelled RydC RNA (Figure 9 panel B, lanes 8-14). We conclude that the low in vivo stability of RydC-S1 is likely due to its lower affinity for Hfq and its relatively weak performance in competition for chaperone binding. Additionally, our competition experiments suggest that RydC wild type and S1 mutant share at least an overlapping binding site. This result is described in the modified text and included in the revised supplementary Materials and methods section.Author response image 1.(A) Electrophoretic mobility shift assay (EMSA) with in vitro synthesized, 5' end-labelled RydC and RydC-S1 RNAs (RydC* and RydC-S1*, 4 nM) in the presence of increasing concentrations of Hfq protein as indicated. (B) Preformed Hfq/RNA*(RydC* and RydC-S1*, 4 nM) complexes were incubated with increasing concentrations of cold competitor RNA (RydC-S1 for RydC*; RydC for RydC-S1).

Regarding the generality of the RydC/Hfq interaction and its bearing on other sRNA/Hfq complexes, we note that the interactions with the 3’ polyU tail and the UU steps are likely to also occur in the recognition of many other sRNAs ([54]; Sauer et al, 2011).

The reviewer also raises a question about the protection of RydC by Hfq from decay. We assume that the lower stability of RydC-S1 is most likely arising from the susceptibility of the RNA to RNase attack when not bound by Hfq. Wild-type RydC is very stable in vivo (t_1/2_ > 32’), and its half-life cannot further increase in a temperature sensitive RNase E mutant (rne-ts) compared to the control strain in the course of a 32’ experiment. We do observe an even flatter decay rate though. This suggests that RydC is mostly immune to the action of RNase E when bound to Hfq. In contrast, the half-life of the S1 pseudoknot mutant was increased in the absence of RNase E (from ∼3 min to ∼6.5 min), suggesting that the S1 mutant is more susceptible to degradation. However, the S1 mutant is not as stable as wild-type RydC and might be preferentially attacked by other cellular nucleases. Whether this susceptibility is due to an alternative structure or reduced binding capability to Hfq is unclear. We have clarified in the text that both the formation of the pseudoknot and the binding of Hfq may contribute to the in vivo stabilization of RydC.Author response image 2.Stabilities of RydC and RydC-S1 were determined in *Salmonella* rne-ts by Northern blot analysis of total RNA samples withdrawn prior to and at indicated time-points after inhibition of transcription by rifampicin (OD600 of 1). In the temperature-sensitive rne-ts *Salmonella* strain, RNase E retains its normal capacity when grown at the permissive temperature of 28°C. As cells are shifted to 44°C, RNase E undergoes a conformational rearrangement rendering it catalytically inactive (Apirion, 1978; Figueroa-Bossi et al., 2009). To quantify the decay rates, the signal obtained at 0 min was set to 100, and the amount of RNA remaining at each time-point was plotted as a function of time. The time-point at which 50% of RNA had been decayed (dashed line) was calculated to determine the half-life (t_1/2_).

*2)*
Figure 2*: The major issue from the crystallography is that in the crystal, RydC is interacting with two adjacent Hfq hexamers. The authors suggest that the contacts seen on neighboring hexamers can also form on a single one, and that such single interaction is likely to be the functional complex. A clearer discussion of what interacts with each hexamer would at least clarify whether there are mutational tests that might demonstrate the importance of each type of interaction. Related to this point,*
Figure 2
*is the only figure that provides a picture of the whole structure and therefore is important for readers to understand the major conclusions, as well as the nature of the complex with two Hfq hexamers. It might be easier to understand this if the second RydC (at upper right) in 2B were in a different shade, for instance (lighter), and possibly if a schematic were included to make it clear what is going on (what the contacts are on each proximal face). Highlighting in some fashion (maybe box and expand to the side) the rim interaction region is important in trying to understand the implications of this sandwiched structure for the rim interaction. It is hard to read the label in some cases*. *Is the U-rich tail on one hexamer and the 5' end on the other Hfq hexamer? How important is the 5' end for function?*

Concerning the discussion of what interacts with each hexamer in the crystal lattice, the 3’ poly U tail contacts the proximal face of one hexamer, while the 5’ end interacts with the lateral face of a neighbouring Hfq hexamer in the crystal lattice. Also, the UU steps make pseudoequivalent contacts, so that U23/U24 make interactions with the rim portion in one hexamer while the U46/U47 make nearly the same interactions with the same residues in the second hexamer (as shown in Figure 4). We have changed the text and reorganized the figures to make this comparison more clear. To make it more clear which contacts are made by the 3’ end and 5’ end of RydC with the adjacent Hfq hexamers in the crystal, we have combined Figure 2 with 3 and 4C to show expanded views of the contacts (now Figure 3).

We have changed Figure 2 (now Figure 3) as suggested by the reviewer. We have changed the labels to improve clarity.

Regarding the importance of the 5’ end, this carries the seed region that is complementary to the target mRNA and has been shown to be important for regulation of *cfa* in vivo (14).

*3)*
Figure 5*: The data in this figure is difficult to understand. By stability measurements alone, LI and LI/LII should be similarly defective; however, for activity, LI is much more active than LI/LII, and in terms of accumulation of RydC, LI looks better than LII (or the double)*. *Is there an explanation for this?*

Regarding the differences in LI and LII, we suggest the causes are complex in vivo. Synthesis rate may not be a major factor since all the variants are expressed from the same constitutive promoter. Instead, the differences might be due to rates of assembly onto Hfq, rates of target binding, and of decay. Our tentative explanation for this is that the assembly rate of LI is sufficiently high to make enough active regulator whereas the LI/II double mutant may be compromised for both assembly and stability, resulting in an intracellular copy number too low for *cfa* activation. Such effects are difficult to distinguish in vivo and will need the development of a suitable in vitro assay that can separate sRNA decay and activity in a more quantitative fashion. We suggest that LI and LII are both interacting with Hfq, but there might be two alternative binding patterns involving LI or LII. One of the binding options (the one using site LI) is more beneficial for RydC with regard to stability. If interacting via LI RydC is capable of target binding, but also protection by Hfq. If interacting via LII, RydC is still able to interact with *cfa* but is more susceptible to degradation. The LI/LII double mutant does not bind Hfq, and is degraded without ever interacting with the target. We have changed the text to clarify the limitations in interpreting these data.

*4)*
Figure 6*: Why was an Hfq mutant with 6 changes used for this experiment? It is difficult to know which rim interactions are in fact important, given this drastic change*. *Is this active for RydC regulation of* cfa *in vivo? Is this mutant derivative present at wild type levels in the cell? Does the lateral rim mutant show wild type binding to* cfa *in the absence of RydC and reduced binding in the presence of RydC in vivo?*

We did try individual point mutations, but we couldn’t see any effects on RydC or *cfa* binding using EMSA. We have not explored if this mutant is present at wild-type levels in cells, or if it is active for RydC regulation of *cfa* in vivo. The binding experiments in vivo are difficult to evaluate in a quantitative manner. Both wild type Hfq and the mutant seem to bind *cfa* in a similar way but with different affinity (see binding assay below, right panel).Author response image 3.Hfq wild-type (Hfq^WT^) and mutant (Hfq^Rim^) binding to RydC (left panel) and *cfa* (right panel). 10 nM of 5’ P^32^ labelled RNA was used with 20 nM Hfq and 25 nM polyA or polyU.

*Also, no data is shown on response vs time to obtain the kinetic rates for the SPR experiments in*
Figure 6*. This data and the fit should be added to supplementary information. The response values shown in*
Figure 6
*are low and raise a concern on the accuracy of the evaluated parameters. Additionally, the asymptotic maximum response value for the Hfqrim (1.8) is less than half than that for HfqWT (5.0). If both Hfq's bind* cfa *RNA in a similar way what accounts for this difference? The authors could support their conclusions from the SPR data by including electrophoretic gel shift data on HfqWT and Hfqrim binding to the* cfa *RNA and RydC*.

We have included below representative kinetic profiles from the octet measurements (Figures 12, 13, 14 and 15). There are many profiles (three repetitions for each run) and we do not feel that it would be useful to include these raw data in the revised supplementary Materials and methods section.Author response image 4.Hfq binding to immobilised
*cfa*Author response image 5.Hfq binding to immobilised
*cfa*
in the presence of RydCAuthor response image 6.Hfq^Rim^
binding to immobilised
*cfa*Author response image 7.Hfq^Rim^
binding to immobilised
*cfa*
in the presence of RydC: Traces correspond to different Hfq concentrations: 5 (dark blue), 10 (red), 25 (light blue), 75 (green), 150 (orange), 250 (light purple) and 500 nM (blue). First 400 seconds corresponds to association measurement, next 300 to dissociation.

To continue, we agree that the maximum response values should be similar for the Hfq^WT^ and Hfq^rim^, but we should point out that the measurements are not from an SPR device (such as a biacore), but from interferometry using an Octet instrument, and the response units are proportional to the layer thickness. We have found that these signals differ for different constructs, likely due to differences in the relationship of the various Hfq species and the surface of the sensor. The electrophoretic gel shift mobility data show two bands with titrations for the Hfq^WT^ while only one band is seen for the Hfq^rim^ mutant, and these may account for the different layer thicknesses of the two samples (see Figure 16 below). These data also show that polyU is slightly more effective than polyA for competitive binding of RydC to Hfq, consistent with proximal face binding for both wild type and rim mutants. We also do not observe a strong effect of polyA or polyU on *cfa* binding for both wild-type Hfq and the mutant, what suggests that Hfq^Rim^ interacts with it in a similar manner.Author response image 8.Hfq wild-type (Hfq^WT^) and mutant (Hfq^Rim^) binding to RydC (left panel) and *cfa* (right panel). 10 nM of 5’ P^32^ labelled RNA was used with 20 nM Hfq and 25 nM polyA or polyU.

*5)*
Figure 7*: The authors need to comment on these decay curves. While they emphasize here the very early points (half-life reduced to 5', compared to Hfq null), it is striking that only a fraction of the RNA is in fact degraded (40%?) in this mutant. Does that suggest that a subpopulation becomes sensitive*, *but an almost equal part is stably bound to Hfq? Is this truncated Hfq active for RydC regulation in vivo?*

This is an interesting point, and we agree that our results might reflect that there are two subpopulations of RydC with one being more sensitive to decay in the absence of the Hfq C-terminus (Figure 17 panel A). As suggested by the reviewer, we have investigated *cfa* regulation by RydC in the presence of Hfq70, and found that *cfa* activity was only slightly compromised (Figure 17 panel B). In vitro affinity measurements of Hfq^66^ for RydC show similar binding to the wild type. One potential explanation is that the interaction of RydC with the C-terminus acts as a foothold and increases the on-rate of the RNA. As a consequence there is more free RydC in the cell in the hfq70 background. Bound RydC remains stable and can interact with *cfa* mRNA while the unbound fraction is decayed quickly, leading to a stepwise degradation pattern. We have added a comment to the text about this point.Author response image 9.

*6) Discussion: The authors should comment on the fact that some Hfq homologs do not have this extension as well as name the proteins that have unstructured regions important to other nucleic acid/protein complexes*.

We have added these points to the Discussion.